

# Urban biogeography of fungal endophytes across San Francisco

Emma Gibson and Naupaka B. Zimmerman

Department of Biology, University of San Francisco, San Francisco, CA, United States of America

## ABSTRACT

In natural and agricultural systems, the plant microbiome–the microbial organisms associated with plant tissues and rhizosphere soils–has been shown to have important effects on host physiology and ecology, yet we know little about how these plant-microbe relationships play out in urban environments. Here we characterize the composition of fungal communities associated with living leaves of one of the most common sidewalk trees in the city of San Francisco, California. We focus our efforts on endophytic fungi (asymptomatic microfungi that live inside healthy leaves), which have been shown in other systems to have large ecological effects on the health of their plant hosts. Specifically, we characterized the foliar fungal microbiome of *Metrosideros excelsa* (Myrtaceae) trees growing in a variety of urban environmental conditions. We used high-throughput culturing, PCR, and Sanger sequencing of the internal transcribed spacer nuclear ribosomal DNA (ITS nrDNA) region to quantify the composition and structure of fungal communities growing within healthy leaves of 30 *M. excelsa* trees from six distinct sites, which were selected to capture the range of environmental conditions found within city limits. Sequencing resulted in 854 high-quality ITS sequences. These sequences clustered into 85 Operational Taxonomic Units (97% OTUs). We found that these communities encompass relatively high alpha (within) and beta (between-site) diversity. Because the communities are all from the same host tree species, and located in relatively close geographical proximity to one another, these analyses suggest that urban environmental factors such as heat islands or differences in vegetation or traffic density (and associated air quality) may potentially be influencing the composition of these fungal communities. These biogeographic patterns provide evidence that plant microbiomes in urban environments can be as dynamic and complex as their natural counterparts. As human populations continue to transition out of rural areas and into cities, understanding the factors that shape environmental microbial communities in urban ecosystems stands to become increasingly important.

Corresponding author
Naupaka B. Zimmerman,
nzimmerman@usfca.edu,
naupaka@post.harvard.edu

## INTRODUCTION

Although major cities only cover a small portion of the Earth's total geographic area, more than 50% of the human population lives in these urban centers, and these cities have major impacts on the biogeochemistry, hydrology, and climate of both their immediate surroundings and of the biosphere as a whole (*Schneider, Friedl & Potere, 2009*). With increasing urbanization, understanding the ecology of cities and urban settings has become

critical to human health and well being. Despite their comparatively small geographic size, the high density of human populations in these environments makes them distinct ecosystems with their own unique dynamics (*Sukopp, 1998*). Urban environments represent the convergence of humans from around the world, any plant or animal species those humans might have brought with them, and infrastructures such as roads, sewers, and tall buildings. Despite the complexity that these ecosystems present, they are often overlooked by ecologists because more traditional ecology has focused on 'natural' systems. In urban systems, however, human influence is a primary ecological factor (*McDonnell & Niemelä, 2011*). In recent years, ecologists have begun studying the urban environment just as they would a natural environment, in order to understand the novel environmental conditions this setting presents to the organisms that live there (*Wu, 2014*).

In this study, we focus on the urban ecology of trees in San Francisco, California. Urban trees can play a major role in shaping the ecosystem of a city. Just as the trees in a forest have a considerable impact on its climate and ecology, trees in cities can have notable effects on a city's environment. For example, plant life in cities can impact temperature, air quality, and other aspects of human health (*Willis & Petrokofsky, 2017*). The urban heat island effect, which occurs when 'islands' of heat form as heat gets trapped between tall buildings, is one of the most well-documented unique urban anthropogenic environmental conditions (*Oke, 1973*). Trees in urban environments have been shown to interact with these city-specific environmental factors (*Kong et al., 2014*). For example, trees in cities have been shown to improve urban air quality by taking up significant amounts of carbon dioxide from city air (*Nowak et al., 2014*). As pollution generated in urban centers is one of the major contributing factors to worldwide pollution, trees in urban environments may have a role in managing the environmental impacts of urbanization (*Alberti et al., 2003*).

Due to their importance, there is growing interest in understanding and maintaining sustainable urban ecologies. One potentially major factor influencing health of plants both in nature and cities is the plant microbiome. As the widespread availability of DNA sequencing has made it possible to characterize microbial communities more easily and comprehensively, the microbiome has become a major area of interest in numerous organisms (*Kyrpides, Eloe-Fadrosh & Ivanova, 2016*). Just as the emerging field of human microbiome study has revealed that symbiotic, non-pathogenic microbes can have major impacts on human health (*David et al., 2014*), plants also host numerous symbiotic microbes. Similar to the human microbiome, the plant microbiome contains great diversity and is comprised of multiple distinct communities in various tissue systems such as the roots (rhizosphere), leaf surface (phyllosphere), and leaf interior (endosphere) (*Turner, James & Poole, 2013*). All of these microbiomes can have an impact on their host's physiology. For instance, bacterial root microbes have been shown to play a role in the growth of various plant species (*Gaiero et al., 2013*). They can also play a variety of roles in host physiology, depending on their location within the plant (*Schlaeppi & Bulgarelli, 2015*). These communities can be quite dynamic, and vary with factors such as plant age (*Cavaglieri, Orlando & Etcheverry, 2009*). While each of these sets of interactions is important, here we focus on the microbial ecology of fungal microorganisms living in the leaf endosphere. The endosphere is an ideal system for biogeographic studies, both because

it is highly diverse, well replicated, and because it is a major interface between the host plant and its environment (*Meyer & Leveau, 2012*).

Endophytic microbes are naturally found in the interior of leaves, whether introduced by natural wounds or openings in the leaf surface, or through penetrating the plant surface with hydrolytic enzymes (*Hallmann et al., 1997*). Although some of these microbes may be latent pathogens or decomposers waiting for the leaf to die, others are mutualists that may confer a benefit to their host (*Carroll, 1988*). Endophytes are commonly divided into classes, based on where they are found and what roles they are known to play in host tissues (*Rodriguez et al., 2009*). In wild grasses, Class 1 endophytic fungi (Clavicipitaceous endophytes) have been shown to protect their hosts by discouraging herbivory, and can even affect host reproductive viability in those same systems (*Clay, 1988*). In controlled settings, inoculation experiments with Class 3 endophytes (horizontally transmitted and localized to shoots) have shown that these specific taxa of endophytes can also have an impact on their host's overall health, including factors such as resistance and susceptibility to disease (*Busby, Ridout & Newcombe, 2016*). In nature, Class 3 fungal endophytes can have impacts on their host's physiology, such as limiting pathogen damage (*Arnold et al., 2003*). In this study, we expected to primarily find Class 3 endophytes, which are known for having high diversity both within populations of plants and within plant tissues (*Rodriguez et al., 2009*).

In the wild, endophytic communities display species diversity comparable to that of any macroscopic community, even among individual trees from the same species (*Gazis, Rehner & Chaverri, 2011*). However, determining the factors that influence this diversity across disparate environments remains an area of active research. The biodiversity of these communities can be quite high, especially in areas like tropical forests (*Arnold & Lutzoni, 2007*). In such natural settings, plant-associated microbial community compositions can show clear biogeographic structure (*Andrews & Harris, 2000*). The urban setting, however, may be distinct, because factors such as rainfall and elevation will likely be less apparent over a smaller geographic area, even as new factors such as proximity to roads and tall buildings may introduce effects of their own. Studies of suburban forests in Japan have indicated that an urban setting has a notable impact on endophytic diversity (*Matsumura & Fukuda, 2013*). However, the full impact of urban environmental factors on endophytic communities has yet to be completely understood.

In this study, we used culturing and barcode gene sequencing to identify the species makeup of endophytic communities in the New Zealand Christmas Tree (Māori: pōhutukawa), *Metrosideros excelsa* (Myrtaceae), throughout San Francisco, CA, to quantify the diversity and biogeographic patterns of foliar fungal communities in this urban environment. Based on studies in natural systems, we expected to find high community diversity both within and between sites, as well as a degree of biogeographic structure to these community compositions. We also anticipated that urban environmental factors might play a role in shaping the biogeography in these endophytic communities.

## METHODS

### Host selection

We selected *Metrosideros excelsa* as a focal host tree. *Metrosideros excelsa* is widely planted throughout San Francisco and as such we were able to obtain samples from a large number of trees in a variety of locations throughout the city.

### Site selection

We selected six sites across the city for leaf sample collection (Fig. 1). When selecting sites, we took factors such as traffic levels, estimated temperature, proximity to tall buildings (greater than the 40-foot building-height limit imposed on most residential buildings in San Francisco), elevation, and proximity to the ocean and San Francisco Bay into account. In selecting the sites, we wanted to capture the wide range of potential urban climates. We used street tree location data available from the City of San Francisco (permalink: https://data.sfgov.org/City-Infrastructure/Street-Tree-List/tkzw-k3nq), which documents the location and species of every tree in San Francisco, to choose unique locations around the city with sufficient densities of *Metrosideros excelsa* individuals (Fig. 1).

### Sample collection

We collected small branches from five trees in each of these sites using a clipper pole, collecting at least three sun-exposed outer branches from each tree. Because *M. excelsa* is an evergreen tree and the newer leaves are likely to contain fewer fungi, we controlled for leaf age by only collecting branches which contained dark green leaves that appeared to be at least one year old. We collected all samples on the same day, August 26, 2017, to ensure that daily weather patterns and seasonal effects would not have an impact on the microbial community composition. Once collected, leaves were stored in labeled plastic bags and stored at 4 °C until culturing. All leaves were processed within 48 h of collection. Permission to sample leaves was granted by the San Francisco Bureau of Urban Forestry.

### Culturing methods

From each sample, we selected a subset of dark green asymptomatic leaves for culturing. We surface-sterilized the leaves by first rinsing them with distilled water, then immersing them in a sterile petri dish with 95% ethanol for 10 s, 0.5% NaOCl for 2 min, then 70% ethanol for another 2 min following *Arnold et al. (2007)*. We emptied the dish between rinses and left it closed inside a sterile biosafety cabinet after the last rinse. We then cut the leaves into small (2 mm × 2 mm) pieces with flame-sterilized scissors and placed them into 1.5 mL microcentrifuge slant tubes partially filled with 2% VWR-brand malt extract agar (MEA). For each tree, we prepared six leaves and made 100 tubes, except for the trees from the downtown site. For the trees at this site, we prepared 140 tubes per tree because we found that they had low isolation frequencies in a preliminary sampling bout. All leaves were prepared this way within 48 h of the initial leaf sampling, to prevent death of the leaf tissue from altering the fungal community composition.

After two weeks, we evaluated the tubes for fungal growth and subcultured the emergent fungi from tubes with growth onto 35 mm plates with 2% MEA in order to better evaluate
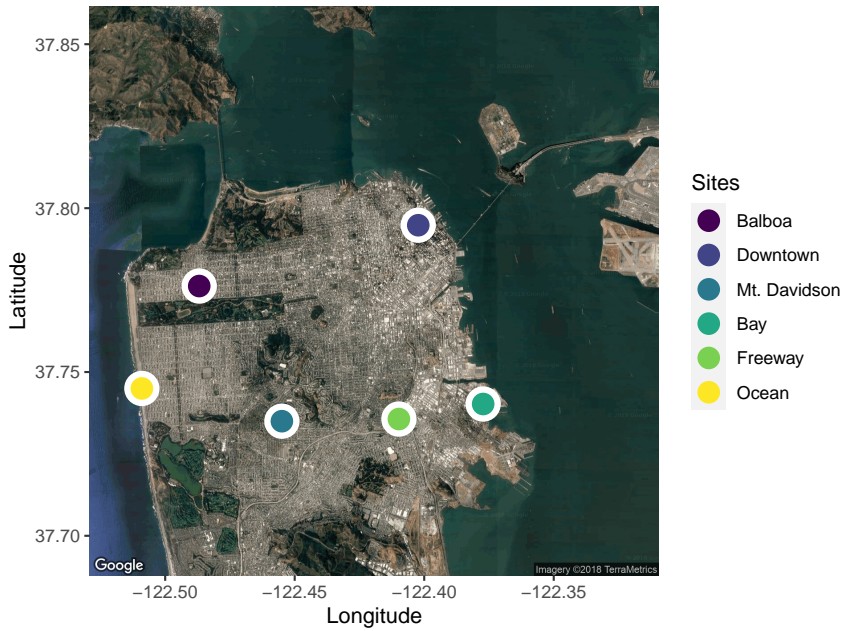

**Figure 1** **A map of the locations sampled across the city of San Francisco, California, USA.** Five trees from each site were sampled. Sites were selected to span the range of environmental conditions across the city. Map data: Google, ©2018 TerraMetrics.

their morphotypes and accumulate sufficient tissue for future barcode gene sequencing and water voucher preparation. We re-evaluated and subcultured these tubes in the following months to capture any late-growing fungi. Each pure culture was vouchered in sterile dI water and stored in a living culture collection in the Zimmerman Lab at the University of San Francisco.

## Molecular methods

We extracted DNA from fungal mycelium using the Sigma RED Extract 'n Amp DNA extraction kit and following previously published protocols (*U'Ren et al., 2012*). First, we added fungal tissue to sterile tubes filled with one mm zirconium oxide beads, then added 100 μL of Extract 'n' Amp DNA extraction solution. Next, we put the tubes in a bead-beater (Mini-Beadbeater-96; BioSpec Products, Inc., Bartlesville, OK, USA) for one minute. The samples were then placed on a heat block at 95 °C for 10 min. After the heating step, we added a dilution buffer to each tube and stored them at 4 °C until PCR.

We performed PCR using fungal-specific primers for the Internal Transcribed Spacer region, a commonly accepted fungal barcode locus (*Schoch et al., 2012*), using the ITS1F forward primer (5′-CTT GGT CAT TTA GAG GAA GTA A-3′) and ITS4 reverse primer (5′-TCC TCC GCT TAT TGA TAT GC-3′). For each PCR reaction, we used 1 μL of template DNA, 10 μL Extract 'n Amp Taq polymerase, 6.4 μL PCR-grade water, 1 μL bovine serum albumin, 0.8 μL ITS1F forward primer, and 0.8 μL ITS4 reverse primer (*U'Ren et al., 2012*). For the PCR reaction, we used a BioRAD T100 thermal cycler with the following cycles: 95 °C for 3 min; 35 cycles of 95 °C for 30 s, 54 °C for 30 s, 72 °C for 30 s;

then 72 °C for 10 min. To ensure that the fungal DNA successfully amplified, and that the master mix was not contaminated, we ran 5 μL of each sample and a negative PCR control on a 1% agarose gel with 1X Tris-acetate-EDTA (TAE) buffer and SYBR Safe. We ran the gel at 120 volts for 20 min and visualized bands using UV transillumination. Successful PCR samples with clean negative controls were kept at 4 °C until sequencing preparation.

To prepare successfully amplified samples for Sanger sequencing, we first cleaned each sample with 1 μL Thermo Fisher Shrimp Alkaline Phosphatase Exonuclease (ExoSAP-IT). To clean the samples, we used the following cycle on a BioRAD T100 thermal cycler: 37 °C for 15 min, 80 °C for 15 min, then hold at 4 °C. After cleaning, samples were kept at 4 °C until they were ready to be sent for sequencing. Directly before being sent for sequencing, cleaned samples that showed bright bands on their gels were diluted with an additional 15 μL PCR water, although a small number of samples that had faint bands on the gels were not diluted. Cleaned samples were sent to MCLabs (South San Francisco, CA, USA) for Sanger sequencing.

## Computational and statistical methods

We used Geneious 11.1 to manually clean and trim the Sanger sequencing data, and to identify and remove failed and low-quality sequences (*Kearse et al., 2012*). Sequences that appeared to have multiple strong signals at a given position were discarded because they may have originated from from a mixed culture. Usable sequences were cleaned by trimming the ends of low-quality nucleotides, clarifying any ambiguity codes, and resolving incorrect base assignment due to dye blobs. Cleaned sequences are available on GenBank (OM974785–OM975639).

After processing the cleaned sequences to add relevent metadata, we used mothur version 1.39.5 to determine Operational Taxonomic Units (OTUs), based on 97% ITS sequence similarity and abundance-based greedy clustering (*Schloss et al., 2009*). Next, we used R 4.1.1 (*R Core Team, 2021*) to analyze the resulting OTU table. We used vegan (*Oksanen et al., 2018*) to: (1) calculate species accumulation curves to determine species richness from the OTU data, (2) estimate unknown total species richness using the Chao estimator with bias correction for small sample sizes (*Chao, 1987*; *Chiu et al., 2014*), (3) calculate Bray-Curtis distances between communities, (4) create a Non-Metric Multidimensional Scaling (NMDS) ordination, and (5) perform a PERMANOVA (*Anderson, 2001*) to assess differences in communities between sites and between trees of different diameters at breast height (DBH) within sites. We used the Tree-Based Alignment Selector (TBAS) toolkit to assign taxonomic context to each ITS sequence (*Carbone et al., 2017*). This toolkit matches unknown ITS sequences to the most similar ITS sequences in a large multi-gene phylogeny of confidently assigned taxa. We used the simple features (*Pebesma, 2018*) and ggmap (*Kahle & Wickham, 2013*) R packages to produce maps of study sites.

Significant differences in fungal leaf isolation frequency and tree diameter were determined using a nonparametric Kruskal–Wallis one-way test of variance and Dunn's Test for multiple comparisons with Holm's correction for multiple hypothesis testing. All analyses, figures, and the final manuscript were generated using a makefile, bash scripts, and R and Rmarkdown. All relevant metadata, as well as analysis and manuscript code, are

available on GitHub: https://github.com/ZimmermanLab/SF-metrosideros-endophytes/ and archived at Zenodo (https://doi.org/10.5281/zenodo.8075450). The complete computational environment for the analyses is documented in a Dockerfile based on rocker containers (*Boettiger, 2014*) and a renv.lock (*Ushey, 2021*) file in that same repository.

### Methods for comparing fungi in San Fancisco trees to New Zealand trees

In order to compare the foliar microbial populations found in *M. excelsa* throughout San Francisco to those in the tree's native range (New Zealand), we surveyed public GenBank records and performed a literature search. First, we searched Google Scholar on July 7th, 2021, for papers mentioning any of the following sets of search terms: "metrosideros endophytes," "metrosideros excelsa endophytes," "metrosideros excelsa foliage fungus," "metrosideros excelsa endophytes." After this search, we looked through the papers cited by those we originally found in order to find any additional studies.

Then, we searched GenBank for fungal ITS sequences that had *M. excelsa* listed as their host organism. We performed this search using the parameters "host ="Metrosideros excelsa", organism =fungi", in order to limit our results to fungal sequences. This search yielded several new sequences that were not included in the previous search because they were unpublished. Using both of these methods, we were able to find about 30 taxa to include as a point of comparison, although many of these were described as pathogenic instead of endophytic.

## RESULTS

Overall, we found high diversity in these urban endophytic communities. From 30 different host trees, we successfully cultured and sequenced 854 fungal isolates, which made up 85 distinct 97% ITS OTUs. Some communities encompassed a low number of fungal isolates; species richness analyses showed that in nearly all of the trees analyzed, our sampling did not encompass the complete microbial diversity (Fig. 2). While we found species diversity within some sites that exceeds 20 distinct fungal OTUs, there appears to be greater species diversity in these fungal endophytic communities than this study was able to document. Chao estimated richness overall was 166.6 with a standard error of 35 and is shown per site in Table 1.

### Isolation frequency and host tree diameter at breast height

The isolation frequency, or the percentage of leaf pieces that yielded fungal isolates, varied significantly between sites (Kruskal Wallis $p < 0.01$, Fig. 3A). In most sites, the isolation frequency also varied between trees, especially in the Bay site. The diameter at breast height (DBH) of trees we sampled from differed significantly as well (Kruskal Wallis $p < 0.01$; Fig. 3B). Trees within the Bay site also show the greatest variation in this measure. However, there was not an overall significant linear relationship between tree DBH and measured isolation frequency (regression $p = 0.076$). The only site that shows consistently low variation in isolation frequency is the Downtown site, which is also the site with the lowest isolation frequencies overall.
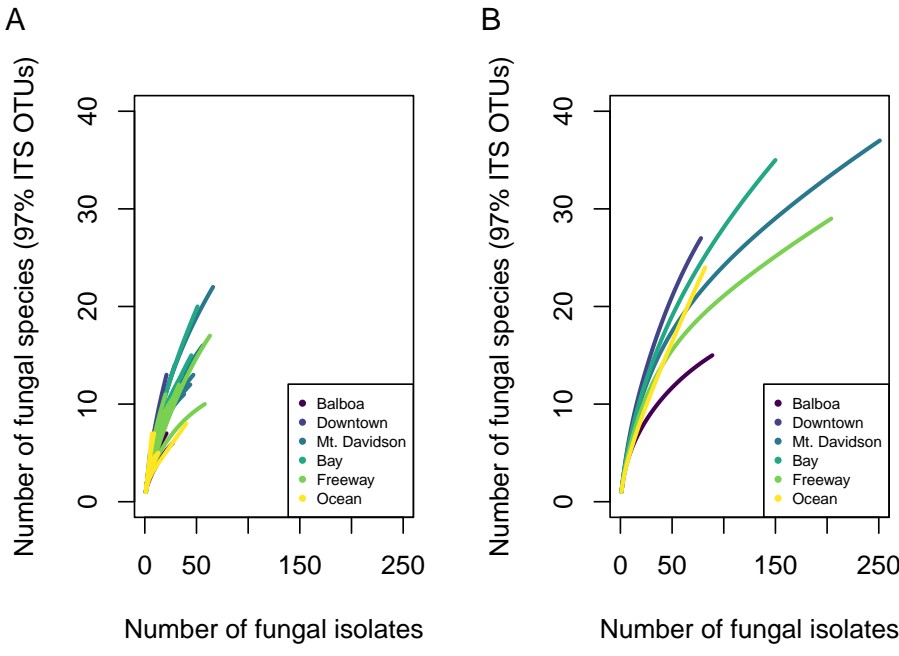

**Figure 2 Species accumulation curves showing fungal community OTU richness.** Each line in A represents the OTU richness for a single host tree. Each line in B represents the combined OTU richness of all trees ( $n = 5$ ) in one site.

**Table 1 Estimated richness of OTU counts across sites using the Chao estimator suggest that OTU richness is higher than observed and is characterized by large uncertainty at several sites.**

| Site | OTU count | Chao estimated richness | Chao richness SE |
|---|---|---|---|
| Balboa | 15 | 19.9 | 5.0 |
| Bay | 35 | 105.5 | 50.9 |
| Downtown | 27 | 44.1 | 11.8 |
| Freeway | 29 | 144.6 | 128.9 |
| Mt. Davidson | 37 | 75.7 | 24.7 |
| Ocean | 24 | 184.0 | 175.7 |

## Diversity patterns

We found 85 total 97% nrITS OTUs among the 30 different trees. Both isolation frequency and the number of fungal species found varied notably between trees. Species accumulation curves showed that the total amount of fungal diversity was not completely sampled in any of the sites (Fig. 2).

The most abundant fungal class that we found in our samples was Dothideomycetes, followed by Sordariomycetes; each had over 300 sequences assigned to them (Table 2). Dothideomycetes was the predominant class in most sites except for the Downtown and Bay sites (Fig. 4). In both of these sites, Sordariomycetes was the most common class. There are several classes that are either absent or present in small numbers in most sites, but more abundant in one or several sites. For example, Eurotiomycetes are more common

A

$\chi^2_{\text{Kruskal-Wallis}}(5) = 18.04$, $p = 0.003$, $\hat{\epsilon}^2_{\text{ordinal}} = 0.62$, $\text{CI}_{95\%}$ [0.59, 1.00], $n_{\text{obs}} = 30$

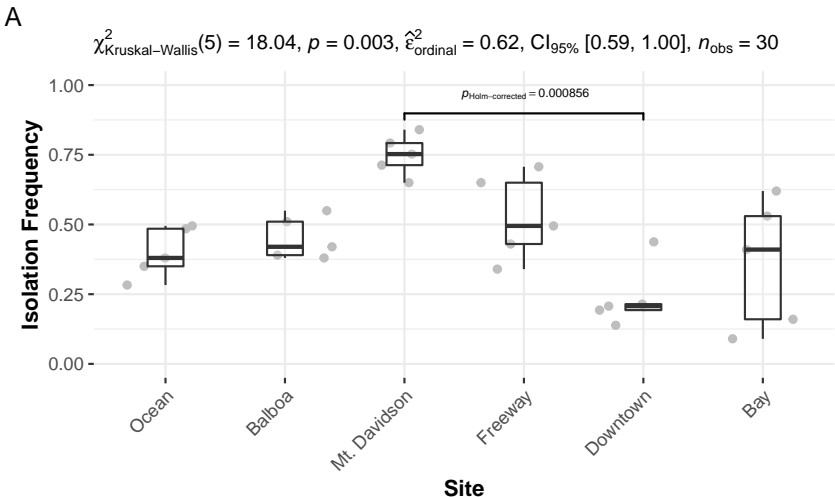

Pairwise test: **Dunn test**; Comparisons shown: **only significant**

B

$\chi^2_{\text{Kruskal-Wallis}}(5) = 22.43$, $p = 4.34\text{e}{-04}$, $\hat{\epsilon}^2_{\text{ordinal}} = 0.77$, $\text{CI}_{95\%}$ [0.71, 1.00], $n_{\text{obs}} = 30$

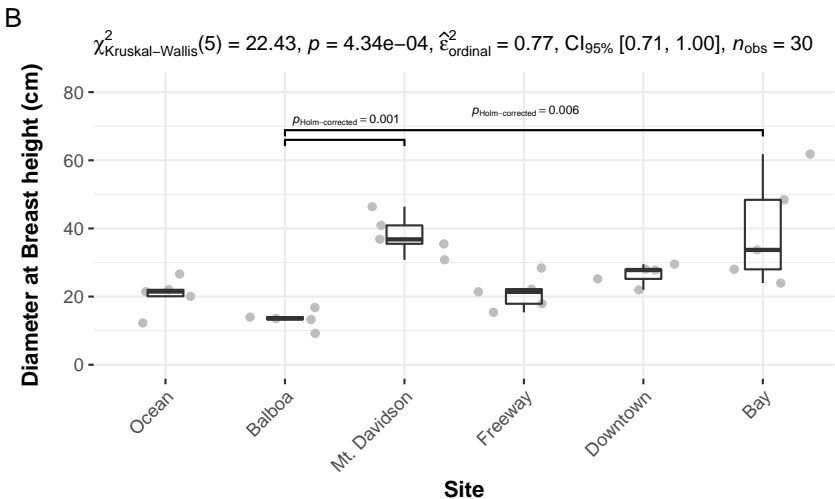

Pairwise test: **Dunn test**; Comparisons shown: **only significant**

**Figure 3 Isolation frequencies (A) and tree diameters (B) at each site.** Isolation frequency (A) is a measure of how many slant tubes showed signs of fungal growth, out of how many total slant tubes were made. Approximately 100 slant tubes were made for each tree except for the trees in the downtown site, which had 140 slant tubes per tree because they had low isolation frequencies during the initial sampling. Sites along x axis are arranged left to right in order of their geographic location from west (left) to east (right).

in the Downtown and Freeway sites, and Leotiomycetes are only abundant in the Mt. Davidson site. Only two sequences were not identified by T-BAS. We queried these two sequences against the NCBI nt database *via* BLASTn and found them to match with high confidence to the Pezizomycetes, based on similarity to a strain recently identified in Australia (*Catcheside, Qaraghuli & Catcheside, 2017*).

**Table 2** **Overall sequenced fungal isolate counts by Class and Order.** All fungi successfully sequenced in this study were placed within the Ascomycota.

| Class | Order | Number of sequences |
| --- | --- | --- |
| Dothideomycetes | Capnodiales | 266 |
| Dothideomycetes | Botryosphaeriales | 127 |
| Dothideomycetes | Pleosporales | 52 |
| Dothideomycetes | unclassified,Capnodiales | 3 |
| Dothideomycetes | Trypetheliales | 1 |
| Dothideomycetes | unclassified | 1 |
| Eurotiomycetes | Eurotiales | 57 |
| Leotiomycetes | Helotiales | 19 |
| Pezizomycetes | Pezizales | 54 |
| Sordariomycetes | Xylariales | 237 |
| Sordariomycetes | Diaporthales | 49 |
| Sordariomycetes | Coniochaetales | 38 |
| Sordariomycetes | Sordariales | 9 |
| Sordariomycetes | Hypocreales | 7 |
| Sordariomycetes | Glomerellales | 1 |

## Biogeographic patterns

While the NMDS ordination group ellipses (standard error around the centroid for each site) suggest that there might be differences in multivariate dispersion among sites, dispersions were not significantly different from one another ($p = 0.76$; Fig. 5), suggesting that PERMANOVA differences are likely to be caused by compositional differences rather than differences in dispersion. Both site and tree DBH were significant predictors of differences in community composition (Table 3), but their interaction was not significant ($p > 0.05$). Sites that were closer together geographically tended to also be closer to one another in NMDS ordination space. We also tested for the effect of tree DBH on community composition differences within sites (*i.e.*, we used 'Site ID' as strata in the adonis function and tested DBH alone as a predictor of community composition differences), and found that it was significant ($R^2 = 0.11$, PERMANOVA $p = 0.006$).

## DISCUSSION

In this study, we sought to characterize the foliar microbiome of *Metrosideros excelsa* trees across the city of San Francisco. We found that the microbial composition of these urban trees' leaves varied in many aspects, from the isolation frequencies of fungi from their leaves to the taxonomic identities of their fungal isolates. Although *M. excelsa's* endophytic communities have been characterized in its native range (New Zealand), there have been few studies about these communities outside of its native environment or in an urban setting (*McKenzie, Buchanan & Johnston, 1999*). In a Hawaiian endemic congener, *Metrosideros polymorpha*, the species makeup of foliar fungal endophyte communities has been shown to vary greatly across gradients of environmental factors such as elevation and rainfall (*Zimmerman & Vitousek, 2012*). While we did not explicitly quantify environmental

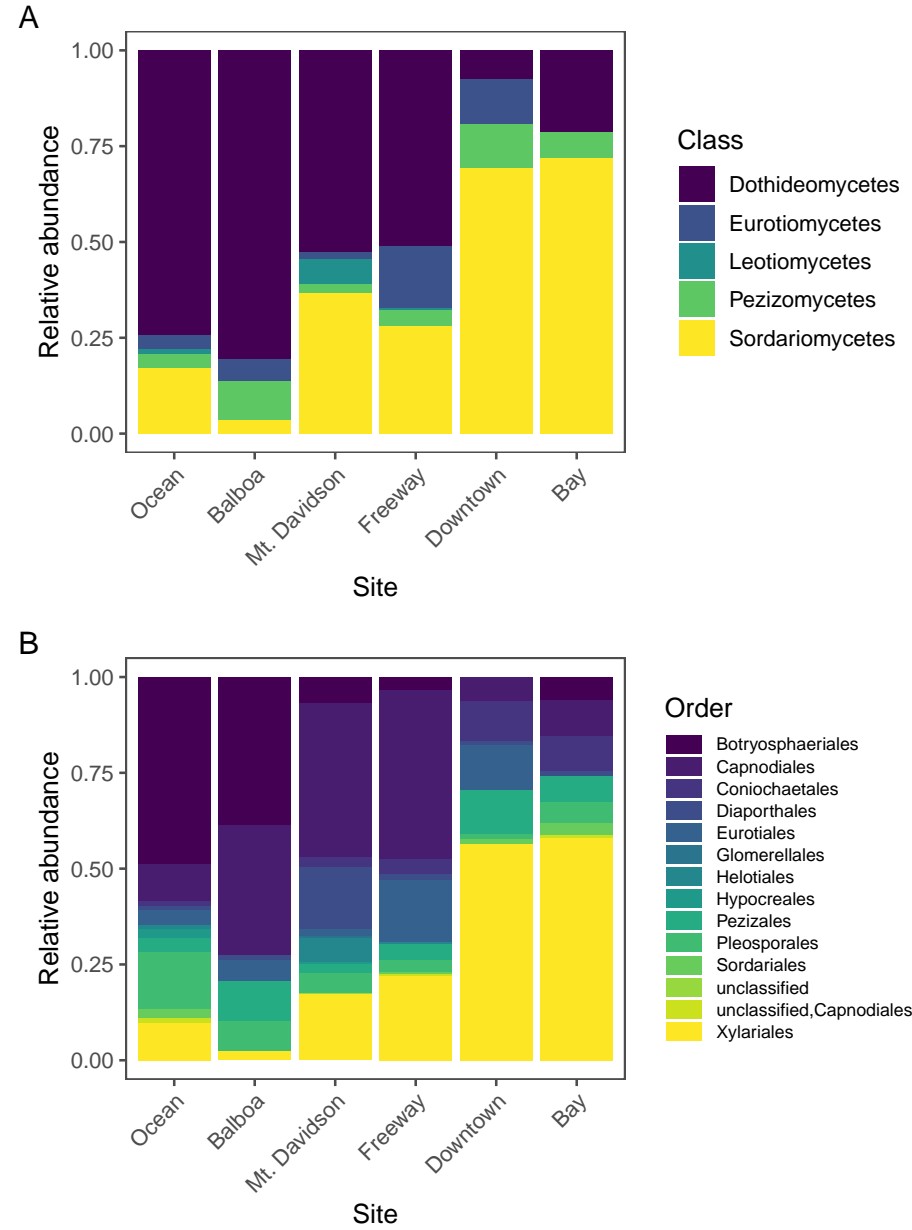

**Figure 4** **Normalized relative abundances of taxonomic groupings (Classes in A and Orders in B) in each site.** Sites along x axis are arranged left to right in order of their geographic location from west (left) to east (right).

variation in this study, the community variation we observed could be explained by either environmental or host physiological factors.

## Isolation frequency and tree size

Both isolation frequency and DBH varied among sites. The trees in the Bay site show the greatest range in both isolation frequency and DBH (Fig. 3). The Mt. Davidson trees have a larger median and range of DBH than trees at the Balboa site (Fig. 3B), and also have a
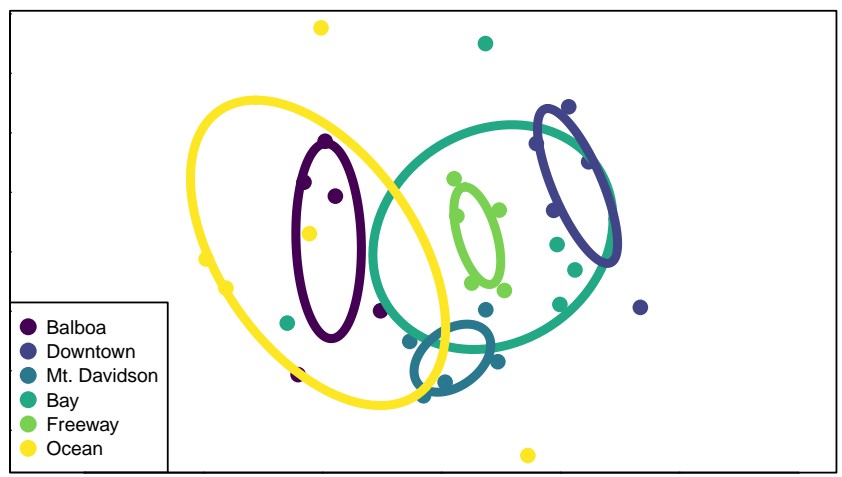

**Figure 5** **NMDS ordination of community compositions.** Each point represents the endophytic community of one tree, and the color of said point corresponds to the site that tree is from. Points that are closer together indicate that the trees they represent have similar community compositions. The ellipses show the standard error around the centroid of all points within a site, and are also color-coded according to which site they represent.

**Table 3** **PERMANOVA model output based on Bray-Curtis distance.** Foliar fungal community composition in leaves of *Metrosideros excelsa* across San Francisco is significantly related to both site of sampling and tree diameter at breast height (DBH), but the interaction between these parameters is not significant.

| Term | df | Sum of squares | Mean of squares | F | $R^2$ | *p*-value |
|------|------|----------------|-----------------|------|-------|-----------|
| Site ID | 5 | 3.642 | 0.728 | 3.015 | 0.373 | 0.001 |
| DBH (cm) | 1 | 0.556 | 0.556 | 2.302 | 0.057 | 0.007 |
| Site × DBH | 5 | 1.208 | 0.242 | 1.000 | 0.124 | 0.484 |
| Residuals | 18 | 4.349 | 0.242 | NA | 0.446 | NA |
| Total | 29 | 9.755 | NA | NA | 1.000 | NA |

significantly higher median isolation frequency than the Downtown site. Just as it shows the greatest range of isolation frequencies and tree sites, the Bay site also has some of the least similarity between its fungal communities on the NMDS ordination (Fig. 5). The trees in the Mt. Davidson site, which have some of the most similar communities (Fig. 5), also have fairly large trees with similar DBH. Although there appears to be a general pattern with larger trees hosting a greater number of fungal endophytes, this realtionship was not significant; thus DBH cannot explain all of the variation in isolation frequencies.

## Diversity patterns

Endophytic microbiomes in tree leaves are known to be highly diverse in natural systems, and that also appears to be the case in these urban trees. The species accumulation curves indicate that even in the most well-represented sites, these are still many potentially undiscovered OTUs within these endophytic communities (Fig. 2). Additionally, there is a considerable amount of taxonomic variation between certain sites, in addition to the generally high diversity of each site (Fig. 4). In such cases, it is likely that environmental

factors play a role in shaping the endophytic communities of these trees. While the impact of environmental factors may be less evident when focusing on fungal endophyte community richness, it becomes more apparent when looking at the identities of these endophytes. The composition of these communities can vary greatly among trees with similar isolation frequencies, as demonstrated by the taxonomic composition of the Mt. Davidson and Bay sites (Fig. 4). It is possible that prevailing onshore weather dynamics, which generally lead the western part of the city to be colder and foggier than the eastern portion, explain some of the patterns we observed. In general, there was higher compositional similarity between trees from the same location than between trees of a similar size (Fig. 5 and Table 3).

## Biogeographic patterns

There appears to be a degree of biogeographic structure to the endophytic communities within these urban trees. Communities of trees from the same site generally cluster closer together than trees from different sites (Fig. 5 and Table 3). Although the Balboa site appears to have a reasonably high isolation frequency (Fig. 3A), many of the slant tubes that showed fungal growth failed to grow into a larger culture, leading to a lower number of usable sequences (Fig. 2B). This low number of sequences per tree may indicate that the divergence in community composition that appears to be present might be due to the low sample size, rather than actual difference between the microbial communities between the Balboa trees (Fig. 5).

However, even sites with low within-site diversity appear to be more similar to other trees from geographically close sites. For example, the Downtown trees do not cluster with the Mt. Davidson, Balboa, or Ocean trees in the ordination, indicating that they have fairly different fungal communities from those sites (Fig. 5). Generally, the westernmost sites (Ocean and Balboa) cluster to one side of the ordination, while the Easternmost sites (Bay and Downtown) cluster to the other, and the sites closer to the middle of the city (Mt. Davidson and Freeway) cluster between them (Fig. 5). Notably, the Downtown and Freeway sites cluster fairly close together (Fig. 5) despite being fairly distant geographically (Fig. 1). In this case, it is possible that there are environmental factors that cause the communities to be more compositionally similar. One of the most likely environmental similarities between these two sites is that they are both exposed to high traffic levels, with one site located in a bustling downtown and the other located near the intersection of two large freeways (280 and 101). Carbon dioxide emissions, particulates, or both of these pollutants could possibly be having an effect on both these trees and their microbiomes. Endophytic bacteria have been shown to have a phytoremediating effect (*Afzal, Khan & Sessitsch, 2014*), so it is possible that endophytic fungi may be playing a similar role in helping these plants persist in high-traffic and thus lower air quality conditions.

## Community composition compared to related trees in other locations

In widely distributed plant species, fungal endophyte composition can differ quite considerably in a species from its native home to the places it has been introduced (*Taylor, Hyde & Jones, 1999*). *M. excelsa* is native to New Zealand, but it and its close relatives in the *Myrtaceae* are widespread across the world. Across all locations, the most abundant

classes found within these *M. excelsa* fungal communities were Dothideomycetes and Sordariomycetes by a notable margin (Table 2). At broad taxonomic levels such as Class, the endophytes found in this study appear to be similar to those found in other Myrtaceae trees elsewhere. For instance, a study of *Myrtaceae* trees in South America, the first and second most common fungal isolates were also identified as Dothideomycetes followed by Sordariomycetes (*Vaz et al., 2014*). Dothideomycetes and Sordariomycetes are also the first and second most abundant classes in the endophytic communities of *M. excelsa's* close relative in Hawai'i, *Metrosideros polymorpha*, as well (*Zimmerman & Vitousek, 2012*). This could indicate that these fungal classes are well-adapted to living as endophytes within these trees, but it is also possible that these communities are actually different when considered at a finer taxonomic resolution.

In order to determine if the fungi we identified were simply members of ubiquitous taxa or if they were potentially more specific to *M. excelsa*, we compared our results to a study of endophytes in two species of trees in the nearby Northern California county of Mendocino. This study looked at two species of host plants: an angiosperm, *Vaccinium ovatum*, and a gymnosperm, *Pinus muricata*. Overall, urban *M. excelsa* shared few of its prominent endophytic classes with *P. muricata*, but it shared Dothideomycetes as a common class with *V. ovatum* (*Oono et al., 2020*). The most common class in *P. muricata* was Atractiellomycetes (*Oono et al., 2020*), which did not appear prominently in any of the trees we studied. Based on this admittedly limited comparison, it appears that angiosperm species may have more endophytic similarities to each other than to gymnosperm species in the same geographic area. However, because we found that the second most common fungal class in our *M. excelsa* leaves, Sordariomycetes, was not present in large quantities in any of the trees in the study by Oono and colleagues, finer-scale host species filtering is also likely to be playing a role in shaping the communities that establish in *M. excelsa*. This conclusion has been suggested in other more comprehensive studies of fungal endophyte host plants as well (*U'Ren et al., 2019*; *Darcy et al., 2020*).

## Comparison of trees in San Francisco and New Zealand

When comparing *M. excelsa's* endophytic taxa found in the literature to those that we collected, we found that the taxa between the regions were somewhat similar on an order level, and even more so on a class level. Only one of the orders that was extremely common in New Zealand (Mycosphaerellales) was absent in San Francisco, and only two of the most common orders in San Francisco (Capnodiales and Pezizales) were absent in New Zealand. However, most of the families that are found in New Zealand are absent in San Francisco, and vice versa. We only found four families in common: Mycosphaerellaceae, Glomerellaceae, Nectriaceae, and Helotiaceae. Of these, Mycosphaerellaceae is the only family that is present in New Zealand and was prevalent (over 50 isolates) in San Francisco. The other families that they had in common were only present in very small numbers in San Francisco, and we were unable to find documentation of three most prevalent families in San Francisco (Xylariaceae, Botryosphaeriaceae, and Cladosporiaceae) being isolated from leaves in New Zealand. Although this could be due to the small sample size, it suggests that *M. excelsa's* endophytic populations may be influenced primarily by their environment.
However, additional research into both the endophytic populations of other trees in San Francisco and *M. excelsa* in its native range would be required to draw further conclusions.

It is important to note the difference in methods between how we collected our samples and how others studying *M. excelsa* in New Zealand collected theirs. Whereas we took care to only collect samples from unfallen, asymptomatic leaves, most of the papers referenced in this comparison use either leaf litter, dead leaves, or obviously symptomatic leaves. Therefore, it is reasonable to infer that the data we are comparing our results to is biased towards pathogenic and decomposing fungi, and underrepresents the non-harmful endophytic populations in *M. excelsa's* native range. Additionally, the number of fungal isolates we were able to find using literature and GenBank searches was much lower than the number of samples we collected ourselves, making quantitative comparison unfeasible. However, we think that such comparisons are still worthwhile as a preliminary look into the extent that *M. excelsa* in San Francisco may maintain its native endophytic microbiome.

## CONCLUSIONS

These findings indicate that the endophytic microbiomes of urban trees are complex and diverse, and may show a degree of biogeographic structure that reflects their natural counterparts. Additionally, it is likely that urban environmental factors play a considerable role in shaping the endophytic communities of these trees. This study has demonstrated that the urban endophytic microbiome is highly diverse and has the potential to be structured by urban environmental factors. Our species accumulation curves indicate that the full diversity of these endophytic communities has yet to be sampled (Fig. 2). Even in the small geographic area of San Francisco, we found notable trends in microbiome composition that may vary with geographic factors, host physical factors such as DBH and age, and uniquely urban environmental factors, such as traffic. A combination of environmental factors and perhaps host physiology therefore appear to be the driving force behind the diversity of these microbiomes. While it is difficult to determine, given the data we have, the exact mechanisms that influence the composition of these communities, the amount of species diversity and biogeographic structure we present here indicate that the foliar microbiomes of urban trees may be just as complex and dynamic as those of trees in nature, and are worth of further concerted study.

## ACKNOWLEDGEMENTS

This study was conducted as an undergraduate Biology honors thesis project by EG under the guidance of NZ. Joshua Copeland and Julian Murdzek assisted with culture processing, vouchering, and DNA extractions. Jeff Oda contributed supplies and equipment expertise.

### Funding

This work was supported by funds from the University of San Francisco Biology Department and the USF Faculty Development Fund. The funders had no role in study design, data collection and analysis, decision to publish, or preparation of the manuscript.

### Grant Disclosures

The following grant information was disclosed by the authors:
The University of San Francisco Biology Department and the USF Faculty Development Fund.

### Competing Interests

The authors declare there are no competing interests.

### Author Contributions

- Emma Gibson conceived and designed the experiments, performed the experiments, analyzed the data, prepared figures and/or tables, authored or reviewed drafts of the article, and approved the final draft.
- Naupaka B. Zimmerman conceived and designed the experiments, performed the experiments, analyzed the data, prepared figures and/or tables, authored or reviewed drafts of the article, and approved the final draft.

### Field Study Permissions

The following information was supplied relating to field study approvals (i.e., approving body and any reference numbers):

Nicholas Crawford, Acting Superintendent of the San Francisco Public Works Bureau of Urban Forestry, stated that no permit was required for the work described here.

### Data Availability

The ITS sequences are available at GenBank: OM974785–OM975639.

The code is available at GitHub and Zenodo:

Available at https://github.com/ZimmermanLab/SF-metrosideros-endophytes.

Naupaka Zimmerman, & emmagibson. (2023). ZimmermanLab/SF-metrosideros-endophytes: v3-accepted-at-peerj (v3-accepted-at-peerj). Zenodo. https://doi.org/10.5281/zenodo.8075450

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
