# Peer review of "Urban biogeography of fungal endophytes across San Francisco"

_PeerJ, doi:10.7717/peerj.15454_

## Round 0.1 · original submission · Minor Revisions

Your manuscript has now been reviewed and the referee comments are appended below. You will see that, while they find your work of interest, they have raised points that need to be addressed by a revision.

Reviewer 1 ·

Basic reporting

This paper is well written, clear and concise. The presentation is logical throughout the paper. The introduction and background are thorough with the relevant literature referenced. The Figures are necessary, high quality, and well labelled.

Experimental design

The experimental design is outstanding based on the methods of previous studies of fungal endophytes of woody plants. The research question is interesting and unusual in that little work has been done on fungal endophytes of trees in urban environments. The methods as described are excellent.

Validity of the findings

The data analysis is rigorous especially important given the many variables. The authors have emphasized the most likely factors that may influence the results namely size of tree and proximity to the bay. In spite of their excellent methods the number of fungal endophytes does not come close to reaching the maximum diversity as shown by the diversity curves. It would take considerably greater sampling to do this. This is an issue with most of the studies dealing with fungal diversity, thus is not unexpected.

Additional comments

A few editorial comments:
In the abstract, ln 12, I suggest inserting “living” before leaves.
Ln 31, insert “of” before the Earth’s…
Under Methods, ln 106, I suggest changing “chose to focus on” to “selected” just for stylistic purposes as you use “focal” later in the sentence.
Lns 108-113, these statements do not belong in the Methods. Perhaps better in Discussion.
Ln 136, delete “until”
Ln 158, 193, and 307, with adverbs ending -ly a hyphen is never needed. This is a great rule to know so one doesn’t have to think about this. Never needed.
Ln 212, change “weren’t” to “were not”
Ln 309, delete “family” as Myrtaceae implies family by the ending “aceae”
Ln 318, delete “quite”
Ln 323, Mycosphaerales??? Mispelled? Not sure, this is not a fungal family with which I am familiar. Is this name correct?
Ln 328, delete “very”
Ln 356, I think you mean “influence” not “influencing”

·

Basic reporting

This paper was clearly written and understandable. The authors describe a survey of fungal endophytes in street trees in San Francisco.
Intro and background are sufficient to show context, however, I think a little more discussion of the types of endophytes would help. Here’s a suggestion for organizing it:
- Introduce classes of endophytes (Rodriguez et al 2009). Talk about what class is likely to inhabit your host species. Talk about why composition could differ among sites or plant species (e.g. ambient environmental differences, environmental filtering). Talk about how those differences may affect your host plant (e.g. disease resistance by preventing pathogen infection)
Figures are very nice looking, relevant to the main points, labeled and described adequately.
All raw data is supplied. This is the most reproducible manuscript I have ever reviewed. Very nice work!

Experimental design

The study objectives were to define the diversity and biogeographic patters of foliar fungal communities in an urban environment. The authors make it clear that little is known about fungal endophyte colonization in urban environments. I would like to see the questions better defined. For instance, What about the urban environment do expect to shape the fungal communities? And how do you plan to measure those things? I give a couple of suggestions for being more quantitative in some areas to improve the explanatory power of your statistical tests and figures.

Validity of the findings

Very high open research standards. Again, nice work on this. The conclusions are well stated, but I am still curious if some of the environmental variables could be linked to the community data in a more quantitative way. See specific comments for some suggestions.

Additional comments

Abstract:
Line 10: You define the plant microbiome as microbes associated with plant tissues. While I think that is not wrong, it’s not explicitly inclusive of rhizosphere microbes. I suggest changing to “plant tissues and soils” or something like that
Introduction:
Line 33: What do you mean by “can be seen” worldwide? In what way?
Line 78-79: Note that these are specific type of endophyte not likely to be found in tree leaves. You might want to introduce the classes of endophytes from Rodriguez et al 2009. It might be helpful in describing what type of endophytes you expect to be present in your host species.
Line 101-103: Odd to see a summary of results in the introduction. I recommend removing this and ending the section with your expectations.
Methods:
Line 116: what is tall? Was that defined in some way? If so, is there a way to make this quantitative and explore its effect on community composition? Maybe you could create a definition for tall – and count the number of building that meet that standard in some defined radius around each site? Or count all structures in a defined radius around the site and create an “index of urban-ness” (just made that up) by weighting taller buildings more than shorter or houses. With that data, you could see if “urban-ness” was a predictor of variation in your microbial community data.
Line 197: Really impressive reproducibility here! Nice work!
Results:
You mention often that the sampling effort was not sufficient to show the total diversity given species accumulation curves. You might want to report chao1 diversity estimates in that case.
Discussion:
Line 282: check code calls to figures.
Line 283: Wondering if it would make your points clearer if you presented centroids with 95% confidence interval bars on your ordination rather than standard error ellipses. That would make it much easier to tell whether your groups are actually different from one another. Given your permanova results, maybe they would not be different, but it looks like some groups may differ from others.
Line 318-319: Also possible that these fungi are ubiquitously common. If you can find a reference of a study looking at fungal endophytes in other species in CA and/or NZ, it might give the reader insight into whether these fungal groups are just common endophytes everywhere.
Table 2: Check code for the first two Terms.
Figure 5: I wonder what this would look like as a db-RDA. You could then add the DBH into your model and add vectors to your ordination. Table 2 suggests that DBH matters for community composition, adding the vectors to the ordination would help visualize that. Just a suggestion.

---

## Round 0.2 · accepted · Accept

I am pleased to see that the authors have revised the manuscript according to the reviewer's comments, and suggestions. I appreciate all their efforts to improve the paper. Thus, I am pleased to inform you that your work has now been accepted for publication in PeerJ. Congratulations!

·

Basic reporting

Excellent work. The manuscript is interesting and well-written. The authors did a great job responding to my comments and have added sufficient information to address all of my prior concerns.

Experimental design

.

Validity of the findings

.

---

## Author Rebuttal · Round 0.2

# Response to reviewers

**Urban biogeography of fungal endophytes across San Francisco**

**Journal:** PeerJ

**Authors:** Gibson E, Zimmerman N

**Manuscript ID:** 67266

We are very pleased to have received such positive feedback on our work and are grateful to both reviewers and the editor for their critical evaluation of the paper. Our responses to each of their individual comments are shown in bold font below. Edits made to the manuscript are indicated in blue: additions are underlined and deletions denoted by "...". Note that line numbers refer to the original submission, rather than the revised manuscript.

## Summary

☐ No edits required   ☐ Revised, as suggested   ☐ Revised, with other edits   ☐ No action taken

**Editor**

[Figure]

| 01 |
|----|
|    |

### Reviewer 1

| 01 | 02 | 03 | 04 | 05 | 06 | 07 | 08 | 09 | 10 | 11 | 12 | 13 | 14 | 15 |
|----|----|----|----|----|----|----|----|----|----|----|----|----|----|----|
|    |    |    |    |    |    |    |    |    |    |    |    |    |    |    |

### Reviewer 2

| 01 | 02 | 03 | 04 | 05 | 06 | 07 | 08 | 09 | 10 | 11 | 12 | 13 | 14 | 15 | 16 | 17 | 18 |
|----|----|----|----|----|----|----|----|----|----|----|----|----|----|----|----|----|----|
|    |    |    |    |    |    |    |    |    |    |    |    |    |    |    |    |    |    |

## Response to reviewers

### Editor

E.1 | Your manuscript has now been reviewed and the referee comments are appended below. You will see that, while they find your work of interest, they have raised points that need to be addressed by a revision.

**Response: Thank you for the time taken to review our manuscript. We believe we have addressed all comments and concerns from the reviewers, below.**

**Action:** Revised, as suggested ✓

### Editor

## Reviewer 1

R1.1 | This paper is well written, clear and concise. The presentation is logical throughout the paper. The introduction and background are thorough with the relevant literature referenced. The Figures are necessary, high quality, and well labelled.

**Response: Thank you for the kind feedback.**

**Action:** No edits required ✓

R1.2 | The experimental design is outstanding based on the methods of previous studies of fungal endophytes of woody plants. The research question is interesting and unusual in that little work has been done on fungal endophytes of trees in urban environments. The methods as described are excellent.

**Response: Thank you for the positive feedback.**

**Action:** No edits required ✓

R1.3 | The data analysis is rigorous especially important given the many variables. The authors have emphasized the most likely factors that may influence the results namely size of tree and proximity to the bay. In spite of their excellent methods the number of fungal endophytes does not come close to reaching the maximum diversity as shown by the diversity curves. It would take considerably greater sampling to do this. This is an issue with most of the studies dealing with fungal diversity, thus is not unexpected.

**Response: Thanks, we agree with you. This is one of the common limitations with environmental microbial surveys, especially when using culture and Sanger-based methods. Other studies have shown that estimates of richness by culture/Sanger based approaches and by culture-free high throughput approaches tend to correlate, despite different absolute numbers. For example, see Supplementary Figure 3 in U'Ren et al. 2019.**

**Action:** No edits required ✓

R1.4 | L12. In the abstract, ln 12, I suggest inserting "living" before leaves.

**Response: Changed.**

**Action:** Revised, as suggested ✓
**Edits:** associated with living leaves

R1.5 | L31. Ln 31, insert "of" before the Earth's...

**Response: Thanks for catching that.**

**Action:** Revised, as suggested ✓
**Edits:** a small portion of the Earth's

R1.6 | L106. Under Methods, ln 106, I suggest changing "chose to focus on" to "selected" just for stylistic purposes as you use "focal" later in the sentence.

**Response: Thanks, nice suggestion.**

**Action:** Revised, as suggested ✓

**Edits:** We ... selected Metrosideros excelsa as a focal host tree.

R1.7 | L108-113. Lns 108-113, these statements do not belong in the Methods. Perhaps better in Discussion.

**Response: We agree. These two sentences have been moved to the first paragraph of the Discussion.**

**Action:** Revised, as suggested ✓

R1.8 | L136. Ln 136, delete "until"

**Response: Deleted.**

**Action:** Revised, as suggested ✓
**Edits:** rinses and left it closed ... inside

R1.9 | L158, 193, 307. Ln 158, 193, and 307, with adverbs ending -ly a hyphen is never needed. This is a great rule to know so one doesn't have to think about this. Never needed.

**Response: Good catch.**

**Action:** Revised, as suggested ✓
**Edits:** commonly accepted, confidently assigned, widely distributed

R1.10 | L212. Ln 212, change "weren't" to "were not"

**Response: Thanks, changed.**

**Action:** Revised, as suggested ✓
**Edits:** were not

R1.11 | L309. Ln 309, delete "family" as Myrtaceae implies family by the ending "aceae"

**Response: Thanks, we've removed "family".**

**Action:** Revised, as suggested ✓
**Edits:** relatives in the Myrtaceae ... are

R1.12 | L318. Ln 318, delete "quite"

**Response: Thanks, removed.**

**Action:** Revised, as suggested ✓

**Edits:** communities are actually ... different

R1.13 │ L323. Ln 323, Mycosphaerales??? Mispelled? Not sure, this is not a fungal family with which I am familiar. Is this name correct?

**Response: Thanks for catching that typo! We meant to write "Mycosphaerellales". The text has been updated.**

**Action:** Revised, as suggested ✓

**Edits:** common in New Zealand (Mycosphaerellales) was

R1.14 │ L328. Ln 328, delete "very"

**Response: Removed.**

**Action:** Revised, as suggested ✓

**Edits:** and was ... prevalent (over 50 isolates)

R1.15 │ L356. Ln 356, I think you mean "influence" not "influencing"

**Response: Thanks, we've changed it.**

**Action:** Revised, as suggested ✓

**Edits:** the exact mechanisms that influence the composition

**Reviewer 2**

R2.1 | This paper was clearly written and understandable. The authors describe a survey of fungal endophytes in street trees in San Francisco.

**Response: Thank you for the kind feedback.**

**Action:** No edits required ✓

R2.2 | Intro and background are sufficient to show context, however, I think a little more discussion of the types of endophytes would help. Here's a suggestion for organizing it: - Introduce classes of endophytes (Rodriguez et al 2009). Talk about what class is likely to inhabit your host species. Talk about why composition could differ among sites or plant species (e.g. ambient environmental differences, environmental filtering). Talk about how those differences may affect your host plant (e.g. disease resistance by preventing pathogen infection)

**Response: Thank you for the suggestion. We've revised the paragraph to introduce different classes of endophytes using Rodriguez et al. as suggested and point out what type we expected to find.**

**Action:** Revised, as suggested ✓

**Edits:** Endophytic microbes are naturally found in the interior of leaves, whether introduced by natural wounds or openings in the leaf surface, or through penetrating the plant surface with hydrolytic enzymes (Hallmann et al. , 1997). Although some of these microbes may be latent pathogens or decomposers waiting for the leaf to die, others are mutualists that may confer a benefit to their host (Carroll , 1988). Endophytes are commonly divided into classes, based on where they are found and what roles they are known to play in host tissues (Rodriguez et al., 2009). In wild grasses, Class 1 endophytic fungi (Clavicipitaceous endophytes) have been shown to protect their hosts by discouraging herbivory, and can even affect host reproductive viability in those same systems (Clay , 1988). In controlled settings, inoculation experiments with Class 3 endophytes (horizontally transmitted and localized to shoots) have shown that these specific taxa of endophytes can also have an impact on their host's overall health, including factors such as resistance and susceptibility to disease (Busby, Ridout & Newcombe, ... 2016). In nature, Class 3 fungal ... endophytes can have impacts on their host's physiology, such as limiting pathogen damage (Arnold et al. , 2003). In this study, we expected to primarily find Class 3 endophytes, which are known for having high diversity both within populations of plants and within plant tissues (Rodriguez et al., 2009).

R2.3 | Figures are very nice looking, relevant to the main points, labeled and described adequately.

**Response: Thank you for the positive feedback.**

**Action:** No edits required ✓

R2.4 | All raw data is supplied. This is the most reproducible manuscript I have ever reviewed. Very

nice work!

**Response: Thank you! That means a lot to us.**

**Action:** No edits required ✓

R2.5 | The study objectives were to define the diversity and biogeographic patters of foliar fungal communities in an urban environment. The authors make it clear that little is known about fungal endophyte colonization in urban environments. I would like to see the questions better defined. For instance, What about the urban environment do expect to shape the fungal communities? And how do you plan to measure those things? I give a couple of suggestions for being more quantitative in some areas to improve the explanatory power of your statistical tests and figures.

**Response: Specific suggestions addressed below. We agree that there is a rich potential to explore more types of environmental factors in explaining the patterns we observed, but given the constraints of this study as an honors thesis we were not able to include as many as we might have otherwise.**

**Action:** No edits required ✓

R2.6 | Very high open research standards. Again, nice work on this. The conclusions are well stated, but I am still curious if some of the environmental variables could be linked to the community data in a more quantitative way. See specific comments for some suggestions.

**Response: Specific suggestions were addressed below.**

**Action:** No edits required ✓

R2.7 | L10. You define the plant microbiome as microbes associated with plant tissues. While I think that is not wrong, it's not explicitly inclusive of rhizosphere microbes. I suggest changing to "plant tissues and soils" or something like that

**Response: Thanks for the suggestion. We've added 'rhizosphere soils' as well.**

**Action:** Revised, as suggested ✓
**Edits:** In natural and agricultural systems, the plant microbiome–the microbial organisms associated with plant tissues and rhizosphere soils–has been shown

R2.8 | L33. What do you mean by "can be seen" worldwide? In what way?

**Response: We have reworded the sentence to increase clarity. We agree it was not clear what we meant.**

**Action:** Revised, as suggested ✓
**Edits:** Although major cities ... only cover a small portion of the Earth's total geographic area, more than 50% of the human population lives in these urban centers, and ... these cities have major impacts on the ... biogeochemistry, hydrology, and climate of both their immediate surroundings and of the biosphere as

a whole

**R2.9 | L78.** Note that these are specific type of endophyte not likely to be found in tree leaves. You might want to introduce the classes of endophytes from Rodriguez et al 2009. It might be helpful in describing what type of endophytes you expect to be present in your host species.

**Response: Thank you, we are aware that different classes of endophytes interact with their hosts in different ways. We have clarified that paragraph (as above) in order to make clear that we understand the distinction and to point out what we are expecting to find.**

**Action:** Revised, as suggested ✓

**R2.10 | L101.** Odd to see a summary of results in the introduction. I recommend removing this and ending the section with your expectations.

**Response: We have removed that sentence as suggested.**

**Action:** Revised, as suggested ✓
**Edits:** ...

**R2.11 | L116.** what is tall? Was that defined in some way? If so, is there a way to make this quantitative and explore its effect on community composition? Maybe you could create a definition for tall – and count the number of building that meet that standard in some defined radius around each site? Or count all structures in a defined radius around the site and create an "index of urban-ness" (just made that up) by weighting taller buildings more than shorter or houses. With that data, you could see if "urban-ness" was a predictor of variation in your microbial community data.

**Response: We thank the reviewer for the comments and suggestions and agree that it would be interesting to perform a quantitative analysis that includes building height as a contributing factor. However, we have decided not to include this analysis because we did not include this data in the initial sampling, and there is not an efficient way to gather this data given current circumstances and resources. Additionally, we do not believe that this data would have a major impact on our analysis, because only one of our sample locations (the Downtown location) had notably taller (more than 5 stories) buildings than the others. While the downtown site was surrounded by numerous skyscrapers 10 stories and higher, all other sites were in residential areas, with buildings 5 stories or lower. Therefore, we believe that an intensive look into the heights of the surrounding buildings wouldn't yield much new insight.**

**Action:** Revised + Clarified 'tall' ✓
**Edits:** proximity to tall buildings (greater than the 40-foot building-height limit imposed on most residential buildings in San Francisco), elevation

**R2.12 | L197.** Really impressive reproducibility here! Nice work!

**Response: Thank you! It took a while to make sure everything worked smoothly.**

**Action:** No edits required ✓

R2.13 │ Results:You mention often that the sampling effort was not sufficient to show the total diversity given species accumulation curves. You might want to report chao1 diversity estimates in that case.

**Response: We thank the reviewer for this suggestion. We have correspondingly added an analysis of richness based on the Chao estimator to the first paragraph of the results, and included a related additional sentence in the methods with several citations, as well as have added a new table with Chao estimated richness values per site.**

**Action:** Revised, as suggested ✓

R2.14 │ L282. check code calls to figures.

**Response: Thank you for catching that. We've fixed the typo in the syntax and now both of those references are rendering properly.**

**Action:** Revised, as suggested ✓

R2.15 │ L283. Wondering if it would make your points clearer if you presented centroids with 95% confidence interval bars on your ordination rather than standard error ellipses. That would make it much easier to tell whether your groups are actually different from one another. Given your permanova results, maybe they would not be different, but it looks like some groups may differ from others.

**Response: We thank the reviewer for the suggestion, but we prefer the current combination of visualization and statistical tests.**

**Action:** Text unchanged ✗

R2.16 │ L318. Also possible that these fungi are ubiquitously common. If you can find a reference of a study looking at fungal endophytes in other species in CA and/or NZ, it might give the reader insight into whether these fungal groups are just common endophytes everywhere.

**Response: While it is possible that these fungi are ubiquitously common, studies by other workers have shown that both host and environmental parameters can have strong effects on shaping the composition of Class 3 foliar endophyte communities. We have included a new paragraph that cites several of these studies, including one that examines angiosperm vs gymnosperm hosts in nearby Mendocino County (Oono et al. 2020). Based on this comparison (particularly Table S12), we think that our tree showed fairly novel taxa composition when compared to non-urban trees in a similar area, but there are also some similarities, particularly with plants that have a similar leaf structure.**

**Action:** Revised, as suggested ✓
**Edits:** In order to determine if the fungi we identified were simply members of ubiquitous taxa or if they were potentially more specific to M. excelsa, we compared our results to a study of endophytes

in two species of trees in the nearby Northern California county of Mendocino. This study looked at two species of host plants: an angiosperm, Vaccinium ovatum, and a gymnosperm, Pinus muricata. Overall, urban M. excelsa shared few of its prominent endophytic classes with P. muricata, but it shared Dothideomycetes as a common class with V. ovatum (Oono et al., 2020). The most common class in P. muricata was Atractiellomycetes (Oono et al., 2020), which did not appear prominently in any of the trees we studied. Based on this admittedly limited comparison, it appears that angiosperm species may have more endophytic similarities to each other than to gymnosperm species in the same geographic area. However, because we found that the second most common fungal class in our M. excelsa leaves, Sordariomycetes, was not present in large quantities in any of the trees in the study by Oono and colleagues, finer-scale host species filtering is also likely to be playing a role in shaping the communities that establish in M. excelsa. This conclusion has been suggested in other more comprehensive studies of fungal endophyte host plants as well (U'Ren et al., 2019; Darcy et al., 2020).

R2.17 | Table 2: Check code for the first two Terms.

**Response: Table 2 has been updated to have properly formatted terms and column headers.**

**Action:** Revised, as suggested ✓

R2.18 | Figure 5: I wonder what this would look like as a db-RDA. You could then add the DBH into your model and add vectors to your ordination. Table 2 suggests that DBH matters for community composition, adding the vectors to the ordination would help visualize that. Just a suggestion.

**Response: While we thank the reviewer for the suggestion, we don't think that it would add much to add a vector for DBH to the ordination, since the amount of variance it explains is very small relative to site, as shown in Table 2. Since we are using NMDS for our ordination visualization, which is an unconstrained method, it does not take into account any of the predictor variables when placing points in ordination space. We feel that the combination of this visualization approach and the statistical interpretation enabled by the PERMANOVA makes the important points that we discuss in our results.**

**Action:** Text unchanged ✗

**References**

Oono, R., D. Black, E. Slessarev, B. Sickler, A. Strom, and A. Apigo. 2020. Species diversity of fungal endophytes across a stress gradient for plants. New Phytol 228:210-225.

U'Ren, J. M., F. Lutzoni, J. Miadlikowska, N. B. Zimmerman, I. Carbone, G. May, and A. E. Arnold. 2019. Host availability drives distributions of fungal endophytes in the imperilled boreal realm. Nat Ecol Evol 3:1430-1437.